# Too-Much-Of-A-Good-Thing Effect of External Resource Investment—A Study on the Moderating Effect of Psychological Capital on the Contribution of Social Support to Work Engagement

**DOI:** 10.3390/ijerph17020437

**Published:** 2020-01-09

**Authors:** Yiheng Xi, Yangyang Xu, Ying Wang

**Affiliations:** 1School of labor and human resources, Renmin University of China, Beijing 100872, China; Yiheng.Xi@vub.be; 2School of public administration, Yan’an University, Shanxi 716000, China; yangyxu@ruc.edu.cn; 3College of economics and management, Nanjing University of Aeronautics and Astronautics, Nanjing 211106, China

**Keywords:** self-determination theory, social support, psychological capital, work engagement, job demands-resources model

## Abstract

Built on the job demands-resources model (JD-R) and self-determination theory, the present research proposed that the relationship between work resources (social support) and employees’ work engagement takes on an inverted U-shaped curve, and presents a model of the moderation of personal resources (psychological capital) on the relationship. The hypotheses were tested by hierarchical regression analysis and path analysis with 535 surveys collected in 19 enterprises. The findings demonstrated an inverted U-shaped curve relationship between enterprises’ social support and employees’ work engagement and further suggested that the predicting effect of social support on work engagement is influenced by employees’ psychological capital, that is to say, the transformation from social support to work engagement bears higher efficiency in employees with high psychological capital than in those with low psychological capital. However, psychological capital fails to display a moderating effect on the curve relationship between social support and work engagement. The present study, casting doubt on the assumption that enterprise supply must meet the needs of employees, argued that the effectiveness of enterprises’ resource support is influenced by the individual needs of employees.

## 1. Introduction

Improvement of the psychological working conditions of employees, as exemplified by work engagement, plays a pivotal role in advancing the efficiency of organizations. Work engagement refers to a kind of work-related affective and cognitive state that is positive, active, and satisfying [1,2]. Numerous studies have shown that employees’ work engagement is a positive predictor of organizational commitment, organizational citizenship behavior, and job performance [3,4] and, as a result, helps achieve higher levels of customer satisfaction of organizations and boost corporate profits [5,6].

With regard to the probe into the antecedents of work engagement, the job demands-resources model [7,8] is the most widely used theoretical model, which suggests a trade-off between resources and demands [1,9]. However, the existing research has found that resources have negative effects in some cases, as exemplified by the decline of the long-term performance in countries that are highly dependent on natural resource revenues (the “resource curse” effect of natural resources [10]). Also, resources that directly create high productivity often place a limit on the company’s ability to cope with market transformation and to explore new opportunities [11,12]. Similarly, is it a given fact that employees possessing abundant work resources will necessarily be more engaged in their work?

It is implied that employees’ work engagement should be founded on the balance between job demands and work resources [13], rather than on the extravagant supply of work resources. The traditional job demands-resources model gives prominence to how resource investment comes into effect in alleviating pressure damage and stimulating employee motivation when there is a lack of work resources and an excess of work demands. However, whether the two processes still occur when there is no shortage of work resources needs to be further explored, as in the present study. A few studies have started to raise questions about the traditional job demands-resources model and claim that excessive work resources or excessively low job demands may not be able to fulfill individuals’ internal needs, thus causing them to lose their enthusiasm for work. In this case, the incentive effect of resources is almost ineffective [14].

As the most fundamental component of work resources, social support also finds itself in a dilemma of incentives as such. According to the too-much-of-a-good-thing theory, favorable resources, such as social support, when in excess, will in turn cause harm. It hints at the potential destruction of work engagement by social support, which has been scarcely studied so far. The current literature predominantly focuses on describing and verifying the assumption that social support should meet the needs of employees (should be doing) but falls short of exploring whether social support can meet the needs of employees under specific circumstances (could be doing) [15,16,17,18]. So, it is difficult to highlight the crux of the too-much-of-a-good-thing effect of resources. Self-determination theory (SDT), known as an antidote to the dilemma, is a process theory that examines into the ways external incentives are ultimately internalized as employees’ internal motivation [19]. According to SDT, the incentives of social support depend on “whether the needs of employees are met or not”. The study, resting on the integration of the job demands-resources model and self-determination theory, focuses on the theoretical issues involved in the too-much-of-a-good-thing effect of resources in social support incentives. In other words, more social support is not always better, and it shares a non-linear relationship with employees’ work engagement.

Besides revealing the “double-sided” effect of work resources (social support) on work engagement, it directs its attention to the promoting or reversing effects of personal resources (psychological capital) on the process. Particularly, when individuals’ motivation is restricted by excessive resources, differences in individual psychological resources may play an important role [20]. However, the unique transformation of work resources fueled by personal resources remains underresearched. The study holds that the functional process of external factors (work resources) is strengthened by the internal environment (personal resources or psychological capital), i.e., psychological capital may serve as an important boundary condition for the relationship between social support and work engagement.

The present study, using empirical investigations, aimed to demonstrate the inverted U-shaped relationship between social support and work engagement and the moderating effect of psychological capital on the relationship. It intends to reveal the “two-sidedness” of the specific work resource–social support and highlight the catalytic role of individual differences (psychological capital) in social support incentives.

## 2. Theories and Hypotheses

### 2.1. Double-Sided Effect of Work Resources

In a world of limited resource allocation defined by economics, it holds true that having more resources is always beneficial. The resource-based view stands out as it advocates the use of resources of strategical significance to define a company’s core competitiveness and value [20,21]. These resources, lying at the heart of strategies and organizations, involve both tangible resources, such as assets and equipment, and intangible resources, such as social capital, brand identity, organizational process, and capabilities of developing or acquiring new products [22]. However, enterprises with resource advantages may fall into the trap of over-dependence on resources, holding back their overall development. Countries that are highly dependent on natural resource income (known as “Dutch disease”) may reduce investment in human capital, technology capital, and infrastructure capital, leading to a long-term performance decline (“resource curse” effect of natural resources) [10]. Similarly, high-productivity resources often restrict a company’s ability to cope with market transformation and explore new opportunities [11,12].

This holds true for workplaces, in that the incentive effect of work resources may also be limited under similar circumstances. In an organization, employees being overpaid tend to, in the first instance, reduce their perceptions of task difficulty and salary equity, instead of staying motivated to increase their output [15]. Therefore, individuals in possession of many organizational resources do not necessarily work harder than those who are not. The too-much-of-a-good-thing theory [23] (TMGT) provides a basic explanation for the phenomenon that “excessively advantageous antecedents may cause destructive effects”. Demerouti and Bakker [24] conclude that all positive features, states, and experiences have their costs, and their positive impacts are diminished when they reach a high level. 

A few studies have focused on the potential negative effects of excessive resources on work engagement. First, the provision of excessive resources by organizations can create an unequal relationship between organizations and employees [16]. The imbalance in the exchange relationship will inevitably debilitate recipients’ esteem, which can only be alleviated after paying equivalent returns. Also, extravagant organizational resources will increase the workload and psychological pressure of employees, preventing them from working wholeheartedly [16,17]. In summary, the double-sided effect of work resources determines that greater resource investment is not always better. The study built a curve relationship model in an effort to reveal the “double-sidedness” of resources in the existing research.

### 2.2. An Inverted “U-Shaped” Relationship between Social Support and Work Engagement

#### 2.2.1. The Linear Relationship between Social Support and Work Engagement

The job demands-resources model is a theoretical model for work engagement [3] in which the incentive process originates from work resources coupled with incentive potential. Thus, all the physical and psychological, social and organizational, tangible and intangible work-related resources, such as social support, work autonomy, and performance feedback [25], can be defined as work resources [25,26]. These resources can assist employees in achieving their job goals, responding to job demands, and staying motivated for learning and development, thereby increasing employee engagement [2,27]. Among others, social support seeks to increase the well-being of employees under the rule of reciprocity between organizations and employees [28], establishing itself as the most representative work resource in stimulating work engagement [29]. According to the job demands-resources model, social support in the workplace refers to the physical, affective, informational, and instrumental assistance that employees receive from the workplace [30,31], ranging from tangible support, such as experience imparting and guidance on business skills, assistance in solving difficulties, or task sharing, to intangible support, such as spiritual comfort and encouragement, etc. [25,32]. According to the existing research, the incentive function of social support is signified by both internal and external incentives [8]. In terms of external incentives, social support can help employees achieve higher job goals and performance [8,9]. In terms of internal incentives, on the one hand, social support enables individuals to be confident in and qualified for their work, thereby generating strong internal incentives and a greater sense of control and achievement at work [3]; on the other hand, social support represents loads of input of social emotions, involving the care and concern for individuals. This can meet the basic needs for individuals’ belonging, allowing them to devote more emotional resources to their work [32,33]. Therefore, the availability of resources serves to reshape employees’ perception and motivation [33], and it also requires the mobilization of considerable resources in order to respond to job demands. At this juncture, more social support, as an important work resource, is “better” for real. Simply, social support positively predicts employee engagement, which has been established by a large quantity of previous empirical findings [34,35,36].

In general, social support is regarded as a representative variable of work resources. Efforts have been made to uphold a “single-sided” research perspective from which social support positively influences work engagement. Evidently, it fails to provide the whole picture of the relationship between social support and work engagement. Whether or not social support continues to maintain its incentive function in the case of excessive provision has not been theoretically verified. The job demands-resources model addresses how resource investment comes into effect in alleviating pressure damage and stimulating employee motivation when there is a lack of work resources and an excess of work demands. However, whether these two processes still occur when there is no shortage of work resources needs to be further explored, as in the present study. 

#### 2.2.2. The Curve Relationship between Social Support and Work Engagement

Some scholars have begun to question the traditional view that “social support must generate incentives for employees” and suggest that social support is damaging in a sense. Studies found that immoderate support, such as task-related assistance, may make employees feel less competitive and threatened with self-esteem damage [37,38]; in addition, employees may assume that the support received is established and does not generate in themselves a sense of obligation to return, so that their voice behavior cannot be mobilized [39]. Besides, as a work resource featuring reciprocity, social support will also bring reward pressure to employees, including the extra-role work pressure [40]. However, these studies predominantly argued from the recipients’ point of view that disruptive or negative reactions are caused by the different understandings of employees about social support, including attribution methods, stress resistance, and the value measurement of social exchange, failing to explore it from the suppliers’ viewpoint, in that what is held accountable is the “quantity” of social support that is supposed to be consistent with the psychological needs of employees. In other words, those studies are still built on the assumption that “resources must meet the needs of employees” and neglect the fact that an excessive resource supply will induce damage to the psychological needs of employees, making it hard to explain that social support may produce negative effects that go against the needs of employees.

The incentive effect of social support is based on the marriage between environmental supply (social support) and individual needs. The fulfillment of individual needs affects individual attitudes and behaviors [41,42]. Therefore, the inflection point at which social support ceases to motivate employees’ work engagement can be spotted at the moment when social support just meets individual needs. From the perspective of the fulfillment of employee needs, self-determination theory reveals the process in which the external resources are internalized into the internal motivation of employees, and serves to explain the curve effect of the relationship between social support and work engagement in the study. On the one hand, social support boosts employees’ willingness in their endeavors to accomplish work tasks by fostering a resourceful work environment [43]. For example, greater social support helps employees to complete their tasks, resulting in more qualified employees for the job, and accordingly, greater empowerment is given to employees. Thus, greater fulfillment of employees’ needs for competence and autonomy can be attained [44]. On the other hand, social support comprises not only the physical goods (instrumental functions) other individuals in the organization provide to employees but also the emotional characteristics of the relationships among the parties in the workplace (affective function) [44]. For example, social support delivers a signal of benign exchange between the two parties, thus meeting employees’ needs for relatedness and belonging [26]. Therefore, the key to effective incentives of social support lies in the matching between its supply and the needs of its recipients, involving both physical and affective aspects.

However, it does not mean that more social support is always better. Instead, extravagant social support will inevitably break the “supply–demand” balance. According to self-determination theory, employee motivation originates from the fulfillment of three basic needs: Autonomy, competence, and relatedness [19,45]. When the social environment restricts or even undermines the fulfillment of these basic needs of employees, their work engagement is immediately reduced [44].

First, with regard to autonomy, social support in excess puts employees in a state where they are forced to rely on organizations, which limits employees’ freedom of choice in their work tasks. This negative perception of “being bound” is even stronger when social support is imposed, because it is hard or even impossible for employees to reject such support. For instance, the premise that individuals can voluntarily accept help from others is that the support and guidance are provided in an appropriate manner. Once the help comes in excess, individuals will feel as though they are “forced” to be dependent, and their initiative needs will be impaired [44]. Furthermore, the continuous increase of resources requires employees’ obligations or labor of “equal value”; as a result, the multi-role responsibilities and the passive binding of individual–collective interests undoubtedly impose limits on employees’ work autonomy. The immediate consequence of these negative perceptions is employees’ internal drive being hit [32,33,46], that is, for employees with excessive social support, the fulfillment of their initiative needs is suppressed so that they are no longer willing to increase their work engagement, giving birth to the “too-much-of-a-good-thing” effect of resources.

Second, employees’ judgement about social support comes with self-directed feedback and awareness [43], such as the perception that whether they are competent enough for such a resource supply. Excessive social support delivers signals that those being supported are not qualified for the job and need extra help, thereby limiting or even undermining the fulfillment of their needs for competence, a typical manifestation of which is the damage to self-esteem or self-core evaluation. Deelstra’s [12] study indicates that employees’ response to social support depends on their perception of whether the amount of support provided threatens their needs for esteem. Extravagant social support spoils employees’ self-esteem at work and instead, conveys to employees a message of being less competitive [37]. From this perspective, employees supplied with excessive social support will be less motivated due to the suppression of their needs for competence.

Third, the purpose of social support also involves meeting individuals’ needs for relatedness [47,48]. Social support meets individuals’ needs for belonging by increasing the opportunities for individuals to engage with others and build more intimate relationships in the workplace. In an organization, social support can help employees find the meaning of group belonging. This means that employees’ needs for relatedness are met because of the status ascribed to them by organizations. However, excessive social support leads to an imbalance of employees’ status in the group, which will trigger multiple “side effects”, typically, the sense of coercive obligation to return [49], sense of oppression with which employees are subject to organizational norms [50], and also sense of the role conflict [51]. Even though the social support provided by organizations is originally intended for the enhancement of employees’ belonging, additional and excessive relationship cost will come at the expense of employees’ work engagement.

Based on the above, it is suggested in the present study that a non-monotonous inverted U-shaped relationship exists between social support and work engagement. More specifically, before a certain inflection point, the increase in social support helps employees become more resourceful and motivated in their work, but continuous support does not yield ceaseless “efforts” from employees. In this case, the expansion of investment in social support will unavoidably bring about a waste of resources, and even give rise to the decline of work motivation. Employees sensing low social support feel that they have no obligation to return because they do not “benefit” from organizations. With the increase of invisible care given by organizations and instrumental resources, the basic psychological needs of employees are easily met. So, the increase in social support will boost employees’ work engagement [8,42]. During this phase, even the care and working conditions given by organizations are not sufficient, employees are still willing to make efforts at work owing to their job identities assigned by organizations, such as career development opportunities and basic remuneration packages for survival. When receiving too many social support resources, employees’ corresponding personal abilities and energy resources are relatively scarce. In this situation, the more social support received by these employees, the lower their work engagement will be.

Compared with the group with high or low levels of social support, the group with moderate social support finds themselves in a situation where they “fall short of the best but are better than the worst,” and they boast adequate resources to fulfill their work obligations within their comfort range. Therefore, the matching of the resource supply and internal needs perceived by these employees comes to a “comfort point”, at which the incentive effect reaches its peak. After passing this inflexion point, they become less stimulated by the continuous increase of social support to step out of the “comfort zone”, yet, in the meantime, they perceive less negative emotions than the group with low levels of social support. In short, what they pay equals what they earn. Thus, it enables them to work in a relatively healthy and balanced way. 

Built on an in-depth understanding of the too-much-of-a-good-thing theory, the following hypothesis can be proposed:
**Hypothesis** **1 (H1).**Social support and work engagement exhibit a significant inverted U-shaped relationship.

### 2.3. The Moderating Mechanism of Psychological Capital in the Relationship between Social Support and Work Engagement

Psychological capital is one’s “positive appraisal of circumstances and probability for success based on motivated effort and perseverance” [52]. It refers to individual cognitive ability to positively evaluate the environment and practice self-regulation [53], which can buffer the “double-sided effect” of social support on work engagement. First, it can assist employees in interpreting internal and external social information and influence social interactions, thereby guiding emotional self-regulation and promoting supportive social interactions [54]. Second, beliefs and motivations associated with high psychological capital may help accumulate the resources or capital of other forms [55]. 

#### 2.3.1. Promote the Fulfillment of Employees’ Needs—The Moderation on the Linear Relationship between Social Support and Work Engagement

The job demands-resources model seeks to integrate internal (personal resources) and external factors (work resources), with an emphasis on the impact of the synergy between them on work outcomes [2,3,7,8,10,12] However, the peculiar effects of differences in individuals’ psychological needs on the incentive effect of social support have not been taken into account. Psychological capital is a personal resource at individuals’ disposal, which can influence their response to and interpretation of the outside world [55]. Therefore, in what way psychological capital, as an internal factor, has an impact on the relationship between social support and work engagement is worthy of attention.

Psychological capital is a reflection of the differences in the psychological feelings and behavioral performance of employees in the face of social support of the same nature and intensity. Specifically, employees with high levels of psychological capital tend to embrace attributions that lead to positive effects with the work environment and adhere to work objectives. Therefore, psychological capital has the potential to be a key moderator of the relationship between social support and work engagement. According to the job demands-resources model [27,54] and self-determination theory [22,32,33], psychological capital is a kind of personal resource, which plays a moderating role between the work environment and outcome variable and may even determine the way employees interpret the environment, make plans, and give responses. Thus, when work resources provide good external conditions for the improvement of work engagement, individuals with more personal resources can give fuller play to these resources.

It is argued in this study that the degree to which individuals “benefit” from social support depends on whether or not they recognize or need it. Additionally, psychological capital serves to improve individuals’ value perception of “being helped” and increase the possibility of individuals embracing positive factors of social support, thus accelerating the transformation process of social support to work engagement. Conversely, when individuals possess a low level of psychological capital, even if organizations provide high-value or high-level social support, such as providing job opportunities and double pay, they are less willing to give a positive response.

Besides, previous research is predominantly focused on the buffering effect of psychological capital on the damage mechanism in work health. That is to say, employees with high levels of psychological capital can respond more effectively to harsh conditions and thus reduce the damage of unfavorable work characteristics, such as work pressures [1,5,34]. Moreover, with the advent of a more complex and variable nature and structure of work, employees with high levels of psychological capital are not “passive recipients” or “respondents” to the work environment [56,57]. Psychological capital acts as a proactive self-management capability, and employees’ own initiative and mastery over the work environment may enable work resources to play a greater role. 

In summary, it is held in this paper that psychological capital will enable employees to interpret the social support provided by organizations in a more positive manner and even actively pursue, leverage, and reconstruct the resource advantages brought by social support. Thus, the following hypothesis can be proposed:
**Hypothesis** **2 (H2).**Psychological capital moderates the relationship between social support and work engagement. The more social support received by individuals with high psychological capital, the more work engagement is presented.

#### 2.3.2. Lift the Resource-Imposed Restrictions on Employees’ Needs—The Moderation on the Curve Relationship between Social Support and Work Engagement

The key to the curve relationship between social support and work engagement rests with whether the “quantity” of social support encourages or hinders the fulfillment of employees’ internal needs. This judgment lies not only in “whether or not employees need it” but also “the appropriateness of the provision of social support to employees’ needs [58]. This appropriateness is derived from employees’ interpretation of social support based on their own needs. Additionally, a number of psychological and personal traits can influence social interaction by helping individuals in their interpretation of social cues, both internal and external, so as to guide their self-regulation and transformation processes [59]. Among these traits is psychological capital, which raises individuals’ positive cognitive assessment of the events in the past, present, and future [60].

Employees can be more motivated by social support when they have a high level of psychological capital. Psychological capital is the psychological ability of employees to transform “supersaturated” resources in an effective way. More specifically, individuals with high psychological capital embrace a positive mindset towards the achievement of established goals (optimism), confident in and capable of executing behaviors to produce performance attainments (self-efficacy), skillful at finding a reasonable way to achieve goals (hope), and tough in the face of adversity (resilience) [61,62]. Therefore, when the basic needs of employees are suppressed by excessive social support, psychological capital strengthens individuals’ perception that work resources are “instrumental”, rather than “stressful”, enabling individuals to think well of social support. For example, employees with high self-efficacy boast greater psychological control over the social support [63]; highly resilient employees are better able to respond to the high expectations of organizations [61,64]; highly optimistic employees embrace attributions that lead to positive effects with excessive social support [65]; and employees with high hope are skillful at finding solutions when restricted by excessive social support [66]. Therefore, psychological capital plays a pivotal role in the way individuals interpret the restrictions imposed by work resources, and leverage the existing resources to advantage [67]. It even equips individuals with the ability to select, change, and leverage resources to achieve their personal goals in light of the conditions and the environment they find themselves in [68]. In sum, psychological capital has the potential to moderate the curve relationship between social support and work engagement. 

Specifically, high levels of psychological capital can even change the direction of the influence of social support on work engagement, the trajectory of which is also “centered” on the mechanism for satisfying individual needs. First, employees with high levels of psychological capital show favoritism to the absorption, recognition, and transformation of work support provided by organizations. Even if social support is in excess, these employees will try to redefine and explain it from a more positive perspective, making the resource supply more attractive and worthy of the investment of time and energy to accept and digest [69], reducing psychological conflicts. In other words, for employees with high levels of psychological capital, excessive social support may not cause them to perceive it as “overloaded” and “incompatible”, thereby reducing the likelihood of its negative effects. Second, psychological capital provides employees with the initiative, energy, and self-discipline necessary to achieve their goals [62]. So, even if excessive social support exceeds their expectations and puts pressure on them, high psychological capital empowers them to deal with the stressors and maintain balance [68], thereby inhibiting the negative effects of social support on work engagement. Finally, psychological capital can supply additional resource gains to employees. Research shows that individuals with high psychological capital have greater access to channels for broadening their knowledge and information [60]. Even if social support is relatively extravagant, they can still resort to a fuller “resource pool” as an offset against the side effects of resource overload. In short, when psychological capital is relatively high, social support and work engagement exhibit a positive linear relationship. 

When psychological capital is relatively low, on the one hand, individuals fall short of confidence to achieve success and positive perception of the environment [70], making it difficult to wriggle free themselves from the constraints of social support overload on individual autonomy and competence. Research has shown that individuals are more likely to be alert to the external environment and be more sensitive to resource loss when they are low in psychological capital [22,32,33]. Thus, it is difficult for them to transform excessive social support into work motivation. On the other hand, as stated above, social support overload causes damage due to the mismatch between employee needs and enterprise supply, and this negative perception will be amplified by low psychological capital. A recent study by Sweetman and Luthans [52] found that low levels of psychological capital will worsen the negative relationship between psychological contract breakdown (“a mismatched manifestation”) and job performance. In the group of employees with lower psychological capital, an initial increase of social support improved their work engagement, yet when the perceived social support continued to increase and go beyond an individual’s psychological needs, its increase instead inhibited their work engagement.

Hence, the following hypothesis is proposed:
**Hypothesis** **3 (H3).**Psychological capital moderates the relationship between social support and work engagement. When psychological capital is relatively low, it shows a linear relationship; when psychological capital is relatively high, it displays an inverted U-shaped curve relationship.


## 3. Methodology

### 3.1. Participants and Data Collection

A random sampling was drawn out of the population of employees involved in various businesses, including basic business sectors (production, sales, service, and project) and administrative sectors (finance, personnel, office, and logistics), in 19 companies across 3 provinces. Taking into account both business and administration divisions, the sample can be seen as representative. The researchers came in contact with each company and made a clarification on the research purpose of the survey and the commitment to the confidentiality of responses. First, data on social support and psychological capital were gathered with the help of the relevant responsible persons or high-ranking officers who were asked to distribute the pencil-and-paper questionnaires to employees in the main sectors of the companies and to provide information on participants’ number, distribution, and time of answering. After that, research assistants were assigned to the companies for the distribution and collection of the second round of the questionnaire survey, which was only dedicated to work engagement. All responses were collected anonymously, and respondents were allowed to create a number based on certain rules in accordance with some of their personal information (this number cannot have respondents identified, though being unique). Sample matching and archiving were done in accordance with this number. Altogether, 800 questionnaires were distributed and 732 were retrieved, of which 535 were valid paired samples, with an effective rate of 73.09%. Female participants accounted for 45.85% of the total. The mean age of the participants was 32.39 years old (SD = 9.01), ranging from 18 to 60 years. In total, 39.13% of them were unmarried and 65.89% held a junior college degree or above. The mean number of working years was 8.94 (SD = 9.13).

### 3.2. Measurement Tools

#### 3.2.1. Social Support 

A simplified version of the Questionnaire on the Experience and Evaluation of Work (QEEW) was used to measure social support [71]. Participants were asked to indicate questions, for instance, “when I face difficulties at work, I can rely on my immediate superior for help” and “when I need help, I can ask my colleagues for help”, based on a 7-point Likert scale, ranging from “0 = never” to “6 = always”. The internal consistency coefficient for the scale was 0.82.

#### 3.2.2. Psychological Capital

The scale used in the Chinese samples by Luthans [12], the initiator of psychological capital, was adapted in the study, which involved 4 dimensions and 24 items. All the items were placed on a 7-point Likert scale. The internal consistency coefficient for the whole scale was 0.83. To be specific, the internal consistency was 0.72 for the sub-scale “hope”, which included 6 questions, such as “at present, I am on the way to achieve my self-imposed work goals”; 0.79 for “self-efficacy”, including questions, such as “I feel I am able to handle a lot of things simultaneously”; 0.76 for “optimism”, including questions, like “I always see the brighter side at work”; and 0.78 for “resilience”, including questions, like “when I find myself in trouble at work, I can come up with many ideas to get rid of it”.

#### 3.2.3. Work Engagement 

The study used a simplified version of the 9-item Utrecht Work Engagement Scale (UWES), which contained 3 dimensions: Vigor, dedication, and absorption, each with 3 items. The Chinese items used were translated by Gan Yiqun and Zhang Yiwen in their work “Reliability and validity test of the Chinese version of Utrecht work engagement scale” [72], which included questions, such as “I am passionate about my work”. The questionnaire was placed on a 7-point Likert scale, ranging from “0 = never” to “6 = always”. The overall internal consistency coefficient was 0.90, 0.84 for vigor, 0.72 for dedication, and 0.83 for absorption.

## 4. Results

### 4.1. Descriptive Statistical Analysis

The means, standard deviations, reliabilities, and correlations of the principal variables and control variables in the study are reported in Table 1. Based on the zero-sequence correlation, the significance of the correlations between the three principle variables all reached 0.001 (Table 1), which suggested the selected variables were highly explanatory.

The study used a questionnaire survey, with all measurements based on employees’ responses. In a bid to avoid common method bias, as suggested by Podsakoff et al. [73], data were gathered based on two phases and with special attention to anonymity and confidentiality in the phase of data collection. In the phase of data analysis, a Harman one-factor test was performed to run an unrotated principal component factor analysis for all variables. It showed that 10 factors had eigenvalues greater than 1 and the variance explained by the first factor was 36.2%, lower than 40%. Thus, it can be concluded that common method bias is not prominent in the study [74].

### 4.2. Validity Testing

Before testing the hypotheses, CFAs (confirmatory factor analysis) were conducted to test the measurement model at the item level to determine whether the scale items adequately indicated their intended underlying constructs [75,76]. 

The factor structure was tested via the confirmatory factor analysis model of structural equation modeling. The results are shown in Table 2. The fit indexes, both SRMR (standardized root mean square residual) and RMSEA (root mean square error of approximation), of the structural factors were less than the ideal value of 0.080, and all other fit indexes reached the desired level, with both TLI (Tucker-Lewis index) and CFI (comparative fit index) greater than 0.900, and 2 < χ^2^/df < 5. It can be seen that the factors reached a good level of fit.

In addition, the VIF (variance inflation factor) values of the primary explanatory variables were all below 3, which was on average 2.66 for social support and 1.88 for psychological capital, indicating that the multicollinearity problems of the principal variables in the study were not prominent.

### 4.3. Hierarchical Regression Analysis and Hypothesis Testing

In order to test the hypotheses, polynomial regression analysis was adopted [77,78,79]. This analysis involves estimating a quadratic regression model, with work engagement as the dependent variable (Y) and social support (X) and psychological capital (W) as the independent variables in our model, along with three quadratic terms constructed from these measures (social support squared, the product of social support and psychological capital, and the product of social support squared and psychological capital). The full polynomial equation is [77]:Y = b_0_ + b_1_X + b_2_W + b_3_X^2^ + b_4_XW + b_5_X^2^W + e.(1)

Multiple stepwise stratified regression analyses were performed (Table 3). In model 1, demographic variables were added as control variables and social support as an independent variable. The results showed a positive relationship between social support and work engagement (b = 0.40, *p* < 0.001). The variance of work engagement explained by all explanatory variables reached 14%. As illustrated in model 2, social support and work engagement displayed a significant quadratic curve relationship (b = −0.02, *p* < 0.05), which additionally explained 1% of the variance. Additionally, the regression coefficient of the quadratic term of social support tested as negative, indicating the curve relationship takes on an inverted U-shape. Thus, Hypothesis 1 was verified. The results of model 3 showed that the linear relationship between social support and work engagement was moderated by psychological capital (b = 0.14, *p* < 0.01), which additionally explained 27% of the variance. Therefore, Hypothesis 2 was verified. At the fourth stage, the quadratic term of social support and the interaction term of psychological capital were added. The results failed to show a significant moderating effect of psychological capital on the relationship between social support and work engagement. Therefore, Hypothesis 3 was not verified.

### 4.4. Competition Model Testing

In order to test the rationality of the hypothesis model, the study compared it with the alternative model. As reported in Table 4:

(1) The quadratic effect model was significantly better than the non-quadratic effect mode (comparison between model b and model a, Δχ^2^/Δdf = 7.496, *p* < 0.001), which, again, validated Hypothesis 1.

(2) Both the nested model and the hypothesis model were better than the quadratic effect model and the non-quadratic effect model. Therefore, regardless of whether an inverse U-shaped relationship exists or not, the linear moderating effect was confirmed (comparison between model c and model b, Δχ^2^/Δdf = 12.031, *p* < 0.001; comparison of model c with model a, Δχ^2^/Δdf = 3.3906, *p* < 0.001), again, verifying Hypothesis 2.

(3) Compared with other alternative models with theoretical rationality, the hypothesis model and the nested model failed to exhibit a significant change, being equivalent to each other (comparison of model d and model c, Δχ^2^/Δdf = 0.7675, *p* > 0.05). The moderating effect of the inverse U-shaped relationship was not confirmed. Hence, Hypothesis 3 was not supported.

Specifically, the results supported the explanatory effect of the inverted U-shaped curve. The inverted U-shaped curve regression is illustrated in Figure 1, from which the explanatory effect of social support on work engagement approached a linear effect, though with a curved arc. This can be attributed to the reason that in enterprise practice, resources provided by enterprises are still finite, which are not enough for all employees to perceive a resource surplus, especially in labor-intensive enterprises. However, it is confirmed in the study that an excessive supply of resources fails to bring about a continuous increase in work engagement.

Although the moderating effect of the inverted U-shaped curve was not confirmed, the linear moderating effect was fully verified and supported by the model comparisons and regression results. Figure 2 serves to further illustrate the moderating effect at issue. Compared with low-level psychological capital, social support has a stronger positive impact on work engagement under the influence of high-level psychological capital, with a consistent direction and positive simple slope (SH = 0.37, tH (535) = 4.71, *p* < 0.001). That is, the higher individuals’ psychological capital level, the stronger the facilitating effect of social support on work engagement, which supports Hypothesis 2 of the study. According to Aiken et al. [74], when testing the moderating effect of the quadratic curve, if only the coefficient of “the interaction term of the moderating variable and the independent variable” is significant, then the moderating variable only changes the inclination of the curve, without changing its shape; if only the coefficient of the moderating variable and the quadratic term of the independent variable is significant, then the moderating variable only changes the shape of the curve without changing its inclination; and if both coefficients are significant, then the inclination and the shape of the curve are changed simultaneously [80]. As indicated by the results, although the coefficient of the quadratic term of the independent variable and the moderating variable was not significant (b = −0.02, *p* > 0.05), the coefficient of the interaction term of the independent variable (social support) and the moderating variable (psychological capital) was significant (b = 0.13, *p* < 0.01). Hence, the empirical results of the study showed that psychological capital moderated the inclination of the curve relationship between social support and work engagement but did not change the shape of the curve relationship.

## 5. Discussions

### 5.1. Theoretical Contributions

First of all, based on the micro-scale of employees, the study responded to the “resource threat statement” of corporate strategy and organizational level, and verified the too-much-of-a-good-thing theory with supplemented micro-scale empirical evidence. In a world of limited resource allocation defined by economics, having more resources is considered positive. As an important work resource, social support has been extensively studied and the motivational functions of its internal and external incentives have been confirmed, especially on the advance of employees’ work engagement [34,35,36]. However, there are also a few studies that have found potential negative effects of social support on work engagement [15,16,17]. The study, integrating different results of social support and work engagement research, adopted an exploratory approach to verify the “promotion-inflextion-destruction” trajectory of the too-much-of-a-good-thing effect, thus providing a comprehensive account of the mechanism behind social support and work engagement; that is, more support resource investment is not always better.

Second, the study investigated the effectiveness of the resource supply from the perspective of employees’ needs. Although a few studies have found the potential negative effects of social support on work engagement, the hypotheses of which were still based on the perspective that “resources can certainly meet the needs of employees”, while it is argued in this study that excessive resources will put a brake on the fulfillment of employee needs. The study further revealed the underlying logic of the too-much-of-a-good-thing theory and placed an emphasis on the individual needs of employees in the investigation of the supply problem; that is, when employees receive social support, whether the external resources are encouraging or hindering the fulfillment of employee needs (principally needs for independence and needs for competence) will have a profound impact on the motivations and actions of employees. This not only fertilizes the research on the supply–demand balance of resources from different perspectives, i.e., employees and organizations, but also casts light on the exploration of the too-much-of-a-good-thing effect of resources.

Third, the results indicated that the functioning process of external causes (work resources) is restricted by the internal environment (individual resources). Employees’ perception of work resources as a threat or as support is shaped by the personal characteristics of employees (such as psychological capital). Psychological capital will enable employees to interpret the social support provided by organizations more positively, and even actively seek, play, and reconstruct the resource advantages brought by such work support. On the one hand, extensive research, based on the job demands-resources model, has discussed the facilitating effects of work resources and personal resources on work engagement in both theoretical and empirical aspects [1,8]. This study added value to the existing theory by proposing the mechanism behind the moderating effects of personal resources between work resources and work engagement, which features a mutual facilitating effect. On the other hand, combined with the thought in positive psychology that “employees are not passive recipients or respondents to work environment but the masters”, the study also revealed that employees with psychological capital are more capable of managing and shaping the work environment in a proactive manner, which empowers work resources to play a greater role.

Finally, the moderating effect of psychological capital on the inverted U-shaped curve of social support and work engagement was not verified; that is to say, regardless of the psychological capital level, the incentive effect of external resource investment will take on a diminishing marginal utility, and will not lead to the sustained growth of work engagement. Therefore, high psychological capital does not endlessly transform and internalize the incentive effect of social support. Even so, it can be seen from the results that the increase of psychological capital may delay the arrival of the critical point of the negative outcomes that result from an excessive resource supply, as indicated by the right shift of the inflection point of the inverted U-shaped curve. Thus, it is suggested that although psychological capital can indeed improve the transformation process of social support and work engagement, its interference effect on excessive resources remains limited. In sum, the study took psychological capital as a positive psychological adjustment environment, with an emphasis on its significant role in the process of resource input-output transformation (including linear and non-linear relationships), which sheds new light on the search for boundary conditions that alleviate the too-much-of-a-good-thing effect.

### 5.2. Practical Implications

The findings of the study will redound to the benefit for organizational management based on two considerations. On the one hand, strengthening workplace facilities, giving care and encouragement, honoring and rewarding, and providing training opportunities are all direct and highly effective ways for enterprises to stimulate employees’ work engagement. However, the interplay between employees and organizations is not simply a mechanical process of input and output and needs to be combined with people-oriented soft management. The study revealed the limitations of the organization-based perspective that an increase of the resource supply must promote employees’ engagement and outcomes. It advocates for special attention to be given to employees’ needs and wishes in the provision of resources and more efforts to strengthen the matching between the organizational supply and employee needs, so as to reduce the wastage arising from enterprises’ expenditures on the amelioration of working conditions and management, and keep business operations running in good order. To be specific, four suggestions are put forward. First, enterprises, on the basis of ensuring normal business operations, are advised to provide sufficient support for employees’ work implementations, especially the execution of tasks, such as optimizing the existing workflows, improving working facilities, and creating a comfortable working environment, all of which are conducive to the inspiration of employees’ work engagement and enthusiasm. Second, compared to instrumental support, more social and emotional benefits should be provided by companies to employees. In a company, employees are not merely position holders, but also the ones who feel affection for the company. Humanistic care, as exemplified by “mentoring”, technical guidance, party lecture learning, league construction, and spiritual counseling, can motivate employees to be more engaged at work. Third, it is advised that the formulation of welfare policies is primarily based on opinions about employees’ needs, which can be easily solicited via interviews or the collection of employees’ wish list. Policymakers need to base themselves on actual conditions and employees’ needs and take a bottom-up approach, thus enabling enterprises’ support for employees to fall into place as expected and setting the stage for the full play of employees’ advantages. Fourth, it is strongly urged that enterprises enhance their communication with employees, either formal or informal, in order to have a good publicity of corporate policies and welfare benefits, and in the meantime, to know in time the needs of employees and the adjustments to their psychological expectations of accessing resources, thus providing more thoughts for employees’ understanding and recognition towards organizations, the implementation of enterprises’ regulations and rules, as well as the consistency of work tasks and goals inside the company.

On the other hand, the study found that high psychological capital of employees not only increases the overall stock of enterprises’ resources but also prompts employees to better understand and recognize the social support provided by enterprises. In this case, employees are more able to make a full transformation of the limited resources provided by enterprises and bring the synergistic effect into play, which is of paramount practical significance for enterprises’ policymaking on employment, assignment, cultivation, retention, and stimulation of employees. There is no doubt that employees’ needs are of equal importance to enterprises’ investment of resources or emotional interaction in the stimulation of engagement. Different employees have different levels of needs. Those who are higher in psychological capital, i.e., highly resilient, optimistic, hopeful, and confident, are more likely to accept support from enterprises and turn it into motivation. A complete integration of employees’ needs is underlain by the thought that social support should be appropriately provided on account of employees’ needs. Therefore, enterprises, in the process of recruitment, training, and selection, are advised to lend an ear to the psychological quality of employees and their exact needs. However, the study also shows that it is never a one-stop solution to the “too-much-of-a-good-thing” effect of resources. For enterprises, relying entirely on the improvement of employees’ abilities is just a makeshift arrangement. More attention needs to be paid to how to stimulate employees based on their needs from a service perspective, in order to “suit the remedy to the case”.

### 5.3. Limitations and Prospects

Although credit should be given to the study for its theoretical and experimental values, it is not free from limitations, which need to be addressed.

The first potential criticism could be leveled at Hypothesis 3, the moderating effect of psychological capital on the inverted U-shaped relationship between social support and work engagement, which was not verified. It is suggested that the too-much-of-a-good-thing effect of resources is characterized by universality and that individual differences fall short of reversing it completely, which lends further support to the too-much-of-a-good-thing effect. Future studies should focus on identifying personality traits or environmental interventions that can reverse the “too-much-of-a-good-thing” effect. For example, the complexity and diversity of employees’ work tasks may also play a moderating role. In the case of highly challenging work, employees need to mobilize more energy and resources to fulfill their work goals, so that better functioning social support can be achieved, and the critical point of social support between “scarcity” and “saturation” will be significantly delayed. Therefore, the complexity and diversity of work tasks may have a positive moderating effect.

Although the research design was based on a two-time data collection, which ensured the causal interpretation of the samples via a separate period of time and reduced the interference of common method bias, the large-scale investigation across 19 enterprises in 3 provinces would inevitably cause inconsistent time intervals between the two phases among the enterprises. Future studies are needed that conduct a longitudinal tracking investigation of relevant models with more rigorous research methods. Besides, future studies are badly needed to explore the “too-much-of-a-good-thing” effect with a more rigorous design as many questions have only been answered partially so far in relation to the way the inflection point of the effect changes, the factors by which and the degree to which it is weakened.

## 6. Conclusions

The present study proposed hypotheses regarding the linear and non-linear relationships between social support, psychological capital, and work engagement, and used stratified regression analysis and path analysis to test hypotheses with 535 samples from 19 enterprises across 3 provinces. The results showed that the influence of social support on work engagement exhibits an inverted U-shaped curve relationship, and that psychological capital moderates the linear relationship between social support and work engagement but does not moderate the curve relationship between social support and work engagement.

## Figures and Tables

**Figure 1 ijerph-17-00437-f001:**
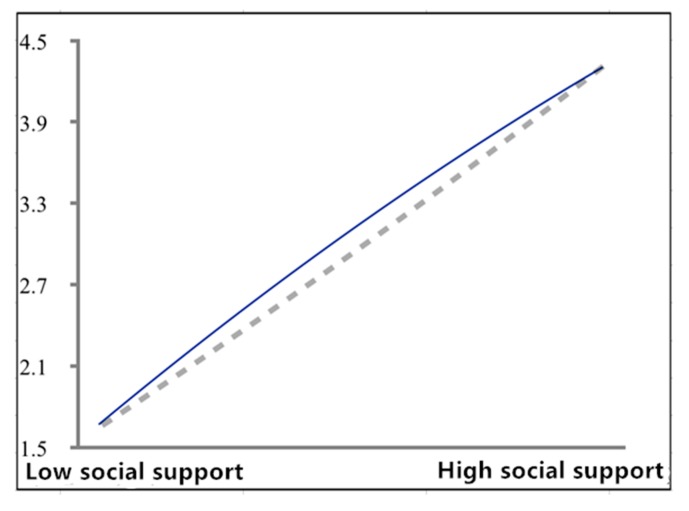
The inverted U-shaped curve relationship of social support and work engagement. Note: The solid line in the figure refers to the inverted U-shaped curve relationship of social support and work engagement. The dotted line in the figure is a straight line crossing the endpoint.

**Figure 2 ijerph-17-00437-f002:**
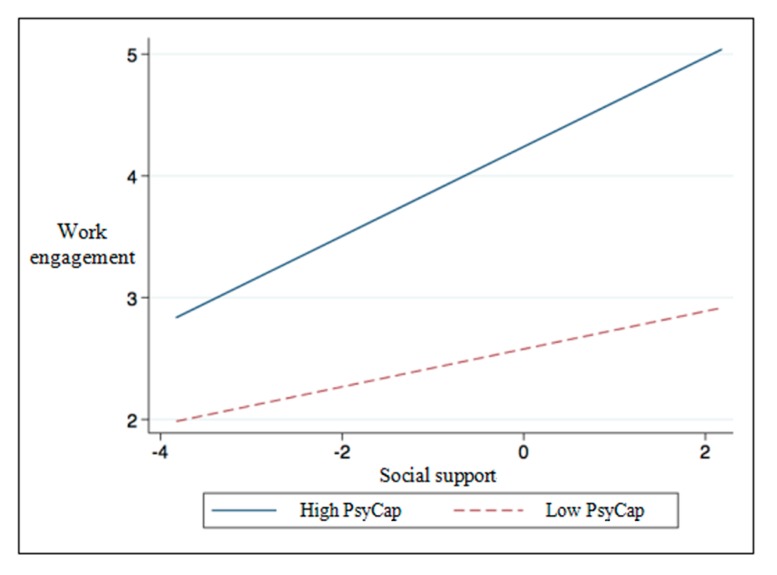
The linear moderating effect of psychological capital on social support and work engagement. Note: The dotted line in the figure is a straight line crossing the endpoint.

**Table 1 ijerph-17-00437-t001:** Means, standard deviations, correlations, and consistency coefficients for each variable.

	Mean	SD	1	2	3	4	5	6	7
1. Social support	3.70	1.27	0.82						
2. Psychological capital	4.17	0.81	0.26 ***	0.83					
3. Work engagement	3.14	1.37	0.22 ***	0.62 ***	0.90				
4. Gender	1.43	0.50	0.00	‒0.09 *	−0.11 *	——			
5. Age	32.39	9.01	0.05	0.12 **	0.09 *	−0.05	——		
6. Marriage	1.63	0.52	0.08 *	0.09 *	0.06	0.00	0.67 ***	——	
7. Educational level	2.36	1.00	−0.00	0.15 ***	0.02	0.05	−0.05	−0.13 **	——

Notes: *N* = 535; *** *p* < 0.001, ** *p* < 0.01, * *p* < 0.05; On the diagonal is the coefficient of consistency. Source: stata software analysis.

**Table 2 ijerph-17-00437-t002:** Fit indexes of confirmatory factor analysis.

Variables	χ^2^/df	*p*	RMSEA (Root Mean Square Error of Approximation)	SRMR (Standardized Root Mean Square Residual)	CFI (Comparative Fit Index)	TLI (Tucker-Lewis Index)
Social support	4.521	0.011	0.078	0.013	0.993	0.979
Psychological capital	2.415	0.000	0.050	0.030	0.972	0.964
Work engagement	4.456	0.000	0.078	0.017	0.990	0.976

Source: stata software analysis.

**Table 3 ijerph-17-00437-t003:** An inverted U-shaped regression relationship between social support and work engagement and the moderating effect of psychological capital.

Demographic Variables	Model 1	Model 2	Model 3	Model 4
Gender (male)	−0.24 *	−0.23 *	−0.11	−0.11
Age	0.01	0.01	0.00	0.00
Marriage (single)				
Married	0.01	0.03	−0.01	−0.00
Divorced	−0.46	−0.33	−0.38	−0.36
Education level (junior college or below)				
Bachelor	0.04	0.03	−0.22 *	−0.22 *
Master or above	0.27	0.30	0.08	0.07
Independent variables				
Social support	0.40 ***	0.53 ***	0.25 ***	0.26 ***
Social support*social support		−0.02 *	−0.01 a	−0.01 a
Moderating variables				
Psychological capital			1.02 ***	1.05 ***
Interaction term				
Social support*psychological capital			0.14 **	0.13 **
(Social support*social support)*Psychological capital				−0.02
Constant	2.82	3.11	3.40	3.39
F value	12.08 ***	11.21 ***	37.58 ***	34.16 ***
Adjusted R-squared	0.13	0.13	0.41	0.41
Change in R Square	0.14	0.01	0.27	0.00

Notes: *N* = 535; the dependent variable is work engagement, *N* = 535, a *p* < 0.1, * *p* < 0.05, ** *p* < 0.01, *** *p* < 0.001. Source: stata software analysis.

**Table 4 ijerph-17-00437-t004:** Fit indexes of confirmatory factor analysis.

Model	χ^2^/df	RMSEA	CFI	TLI	Δχ^2^ (Δdf)	Notes
a. Non-quadratic effect model	38.243	0.121	0.977	0.943	38.243(4) ***	
b. Quadratic effect model	67.227	0.113	0.971	0.945	29.984(4) ***	Compared to model a
c. Nested model	55.196	0.094	0.978	0.948	12.031(1) ***	Compared to model b
d. Hypothesis model	56.731	0.084	0.983	0.956	1.535(2)	Compared to model c

Notes: Model a is the non-quadratic effect model, which the quadratic term of social support was not put into; model b is the quadratic effect model, which the quadratic term of social support was put into; model c is the nested model, which the product term of psychological capital and social support, and the quadratic term of social support were put into; model d is the hypothesis model. *** *p* < 0.001. Source: stata software analysis.

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
