# Peer review of "Too-Much-Of-A-Good-Thing Effect of External Resource Investment—A Study on the Moderating Effect of Psychological Capital on the Contribution of Social Support to Work Engagement"

_ijerph, 2020, doi:10.3390/ijerph17020437_

Round 1

Reviewer 1 Report

see attached file

Reviewer 2 Report

Review for International Journal of Environmental Research and Public Health

Comments for Authors

Manuscript ID: ijerph-669346

Title: Too-Much-of-A-Good-Thing Effect of External Resource Investment. A Study on the Moderating Effect of Psychological Capital on the Contribution of Social Support to Work Engagement

This paper examines data from a survey of employees in 19 companies in china to examine the relationship between work resources and work engagement.

This is an ambitious study that provides some complex data. The paper, whilst interesting, is too long and torturous. It would benefit from a tighter focus on the aims, hypotheses, methodology and results. A shorter, focused paper would be preferable.

Specific points are:

Introduction – much of this is repeated in section 2 (Theories and Hypotheses). The introduction should be short and to the point focusing on the theoretical background and presenting a short paragraph on the aims of the study. Theories and Hypotheses. This should be shortened to clearly state and justify the study hypotheses. These hypotheses should form the focus of the results section. More information is required on the sampling and data collection. Why did the authors select the 19 companies, how was this done, what are these companies’ representative of? How did they actually collect their data? They need to justify their selection of companies and participants and thus show how generalisable their results are. The data on the individuals in their sample should be placed in the results section. The authors should add a statistical analysis sub-section in the methodology section showing their plans and reasons for the statistical analysis. The results section should provide the descriptive statistics of the sample and then focus on the three hypotheses to show how these are supported (or not) by the analysis.

Reviewer 3 Report

Too-Much-of-A-Good-Thing Effect of External Resource Investment.------A Study on the Moderating Effect of Psychological Capital on the Contribution of Social Support to Work Engagement

This manuscript is based on  535 surveys collected in 19 enterprises. Aims to check which of the two hypotheses put forward is best fulfilled. It is a rigorous work with a great statistical analysis, sometimes complex to understand. I propose the following suggestions to improve the manuscript

The objective should be more clearly defined in the introduction. Possible conclusions should not appear in the introduction. Remove or rewrite.

E.g. lines 108 to 112 “The conclusions are intended to show that developing and boosting employees' psychological capital are more effective than changing organizational practices and policies in an attempt to improve employee work engagement. The study represents a major supplement to the existing theory and practical guidance for its probe into the antecedents of and interventions for (how to mobilize employee motivation) employees’ work engagement.

I suggest adding the results of the surveys before the statistical analysis. To add an explanatory diagram of the methodology, where the role of hypotheses is highlighted. I suggest writing some brief conclusions separately from the discussion, this way the reader will have a clearer idea of the work done these conclusions should meet the objectives clearly defined in the introduction

Round 2

Reviewer 2 Report

Review for International Journal of Environmental Research and Public Health 

Comments for Authors

Manuscript ID: ijerph-669346 (REVISED)

Title: Too-Much-of-A-Good-Thing Effect of External Resource Investment. A Study on the Moderating Effect of Psychological Capital on the Contribution of Social Support to Work Engagement

I previously reviewed this paper. The authors have tried to address the reviewers’ comments and have re-submitted an improved paper.

Reviewer 3 Report

The authors have correctly added all the suggestions to this new version.

From my point of view the manuscript is acceptable for possible publication. 

Congratulations to the authors for this work

This manuscript is a resubmission of an earlier submission. The following is a list of the peer review reports and author responses from that submission.